

**Where does the dust deposited over the Sierra Nevada snow come from?**
Huilin Huang[1*], Yun Qian[1*], Ye Liu[1], Cenlin He[2], Jianyu Zheng[3,4], Zhibo Zhang[3,4], Antonis Gkikas[5]
[1]Atmospheric Sciences and Global Change Division, Pacific Northwest National Laboratory, Richland, WA,
USA
[2]Research Applications Laboratory, National Center for Atmospheric Research, Boulder, CO, USA
[3]Department of Physics, University of Maryland Baltimore County, Baltimore, MD, USA
[4]Joint Center for Earth Systems Technology, University of Maryland Baltimore County, Baltimore, MD, USA
[5]Institute for Astronomy, Astrophysics, Space Applications and Remote Sensing, National Observatory of
Athens, Athens, Greece
*Corresponding to: Huilin Huang, Huilin.huang@pnnl.gov; Yun Qian, Yun.Qian@pnnl.gov





**Abstract**
Mineral dust contributes up to one-half of surface aerosol loading in spring over the southwestern U.S.,
posing an environmental challenge that threatens human health and the ecosystem. Using the self-
organizing map (SOM) analysis, we identify four typical dust transport patterns across the Sierra Nevada,
associated with the mesoscale winds, Sierra-Block-Jets (SBJ), North-Pacific-High (NPH), and long-range
cross-Pacific westerlies, respectively. We find dust emitted from the Central Valley is persistently
transported eastward, while dust from the Mojave Desert and Great Basin influences the Sierra Nevada
during mesoscale transport occurring mostly in the winter and early spring. Asian dust reaching the
mountain range comes either from the west through straight isobars (cross-Pacific transport) or from the
north in the presence of NPH. Extensive dust depositions are found on the west slope of the mountain,
contributed by Central Valley emissions and cross-Pacific remote transport. Especially, the SBJ-related
transport produces deposition through landfalling atmospheric rivers, whose frequency might increase in a
warming climate.



**1. Introduction**
The emission, transport, and deposition of mineral dust (hereafter dust) are processes receiving
increasing interest from the scientific community (Sarangi et al., 2020). Dust emission is an integral part of
aridification and mirrors the effects of climatic change and anthropogenic land use on the dust-prone area
(Duniway et al., 2019). Airborne dust interacts with Earth's climate system by altering radiation budget and
cloud lifetime and amount (Forster et al., 2007; Haywood et al., 2005; Huang et al., 2019). Its adverse
impacts on human health, ranging from cardiovascular illnesses to premature mortality, are well-
documented by numerous epidemiological studies (Laden et al., 2006; Lim et al., 2012; Crooks et al., 2016).
The deposition of dust on snow surface influences snow albedo, further contributing to anthropogenic
climate change as early as the 1970s (Qian et al., 2009; Qian et al., 2014; Skiles et al., 2018).
Dust over the southwestern U.S., particularly in California and Nevada states, is an important
aerosol type contributing to more than half of surface aerosol concentrations in spring (Kim et al., 2021).
Covered by dry soil with large gaps and sparse vegetation, the surrounding Mojave Desert, Sonoran Desert,
and Great basin are susceptible to wind erosion (Okin et al., 2006; Duniway et al., 2019). The dry or
ephemeral lakes in the deserts produce very fine dust containing toxic inorganic constituents (Goldstein et
al., 2017). In addition, anthropogenic land-use practices – e.g., agriculture and human settlement, have
greatly disturbed crustal biomass and produced windblown dust along the west coast (Pappagianis and
Einstein, 1978; Clausnitzer and Singer, 2000; Neff et al., 2008). Furthermore, cross-Pacific dust transported
from Asia and Africa to the Sierra Nevada range is widely reported (Ault et al., 2011; Creamean et al., 2014;
Creamean et al., 2013). The surface dust concentration has been found to increase in the past two decades
during spring at sites across the Southwest (Tong et al., 2017; Hand et al., 2017; Brahney et al., 2013), and
the onset of dust season is shifting earlier in response to climate change (Hand et al., 2016). The elevated
dust emission and earlier dust season are supposed to lead to a spectrum of environmental and societal
impacts in the most populated U.S. state. Especially, the resultant dust deposition on mountain snow
decreases snow albedo and produces a radiation forcing of 0-14.6 W m$^{-2}$ during the melting season (Huang
et al., 2022a), shifting snowmelt timing to earlier dates and further increasing California's vulnerability to





water resource fluctuations (Wu et al., 2018; Huang et al., 2022b). With its complex terrains, frequently
varying microclimate, and coexisting sources from both local and remote regions, the Sierra Nevada area
is an interesting region for studying dust transport and its response to climate change.

Characterization of dust emission, transport, and deposition across the Sierra Nevada has been

investigated using various data. Isotopic analyses (i.e., concentrations of Pb, Nd) are widely used to
distinguish and quantify the respective contribution of dust emission from local (dried Owen Lakes),
regional (Central Valley and the Mojave Desert), and global sources (Asia and Africa) on the dust
deposition on the mountain (Muhs et al., 2007; Jardine et al., 2021; Aciego et al., 2017; Aarons et al., 2019).
Their source attribution has been generally confirmed by the analyses of dust particle size and composition
(Creamean et al., 2014; Creamean et al., 2013; Reheis and Kihl, 1995). The isotopic and composition
analyses have been commonly used with back-trajectory modeling to further identify the dust transport
pathway from the source to the deposition location (Vicars and Sickman, 2011; Creamean et al., 2014;
Creamean et al., 2013). Yet, these analyses generally retrieve dust sources in a short time and at a specific
location. Alternatively, ground-based measurement networks were established in the 1990s and provide
long-term trends of dust concentrations and the interannual variability across multiple sites (Hand et al.,
2017; Achakulwisut et al., 2017; Hand et al., 2016). However, they do not contain information on dust
origins and atmospheric conditions responsible for dust transport. Satellite retrievals were less commonly
used to study dust characteristics across the Sierra Nevada (Lei and Wang, 2014), mainly due to the poor
data coverage caused by cloud contamination in the region.

Global and regional climate-chemistry models have been widely used to understand the drives of

the variability of dust and quantify the role of regional and remote transport, filling the gaps in the
observations (Chin et al., 2002; Chin et al., 2007; Kim et al., 2021; Wu et al., 2017). While dust emissions
and transport have been generally studied, there lacks a connection between dust emissions from the source
region and the timing, location, and amount of dust deposition to the Sierra Nevada snow. The isotopic and
composition analyses attribute dust sources at a few sites. But to our knowledge, no regional
characterization has been conducted on how dust is transported to the Sierra Nevada after emissions from



adjacent drylands and remote continents and when, where, and how much depositions occur for dust
transported through different pathways. The connection between dust emissions, transport pathways, and
deposition to snow would facilitate the prediction of future changes in dust regimes and the corresponding
climate impact, enabling more efficient management practices. With a focus on the dust that influences the
Sierra Nevada, this study investigates 1) Where does the dust come from? 2) How is dust transported to the
mountain from the sources? 3) How is the dust deposited on the Sierra Nevada during spring, when the
dust-in-snow largely influences snow albedo and snowmelt? In consideration of both dust emission season
and mountain snow duration, we confine our study period from February to June 2019. We integrate models
and observations to understand how the dust deposition is linked to a specific source both surrounding and
far from the Sierra Nevada.

**2.1 Model and Reanalysis datasets**
**2.1.1 WRF-Chem configuration**
Table 1. Model configuration.

| Atmospheric processes | WRF-Chem Configuration |
| --- | --- |
| Meteorological IC/LBCs | ERA5 |
| Microphysics | Morrison double-moment |
| Radiation | RRTMG for both shortwave/longwave |
| Land surface | CLM4 with SNICAR |
| Surface layer | Revised MM5 Monin-Obukhov |
| Planetary boundary layer | YSU scheme |
| Cumulus | Grell-Freitas |
| Chemical driver | MOZART |
| Aerosol driver | MOSAIC 4-bin |
| Anthropogenic emission | NEI2017 |
| Biogenic emission | MEGAN |
| Biomass burning emission | FINNv2.2 |
| Dust emission | GOCART |
| Chemical IC/BC conditions | CAM-Chem |




We used the WRF-Chem version 3.9 to study dust emission and transport across the Sierra Nevada.
The model setups (Table 1), including the physical schemes and emission inventory, follow Huang et al.
(2022a), which showed that the model captures the distribution and variation in aerosols reasonably well in
the study domain (126.12-112.86°W, 32.3-43.0°N). The Model of Ozone and Related chemical Tracers
(MOZART) chemistry module (Emmons et al., 2020) and the Model for Simulating Aerosol Interactions
and Chemistry with four bins (MOSAIC 4-bin) aerosol model (Zaveri and Peters, 1999) were applied, and
dust emissions were calculated "online" using the GOCART dust scheme (Ginoux et al., 2001). The
meteorological initial and lateral boundary conditions were derived from the ECMWF Reanalysis v5
(ERA5) at 0.25° horizontal resolution and 6 h temporal intervals (Hersbach et al., 2020). Spectral nudging
was employed with a timescale of 6 h above the PBL to reduce the drift between ERA5 reanalysis data and
WRF's internal tendencies (Von Storch et al., 2000). The chemical initial and boundary conditions were
provided by CAM-Chem (Buchholz et al., 2019).
We applied the model to two nested domains (Fig. 1). Domain 1 (126.12-112.86°W, 32.3-43.0°N)
was configured to cover all of California, Nevada, and part of the surrounding states with $110 \times 120$ grid
cells at 10 km $\times$ 10 km horizontal resolution; the nested domain 2 covered the Sierra Nevada and
surrounding regions with a 2 km $\times$ 2 km resolution. The cumulus scheme is turned off in domain 2 with
convection-permitting resolution. We used 35 vertical model layers from the surface to 10 hPa with denser
layers at lower altitudes to resolve the PBL. The simulation period ranged from September 20, 2018, to
August 31, 2019.



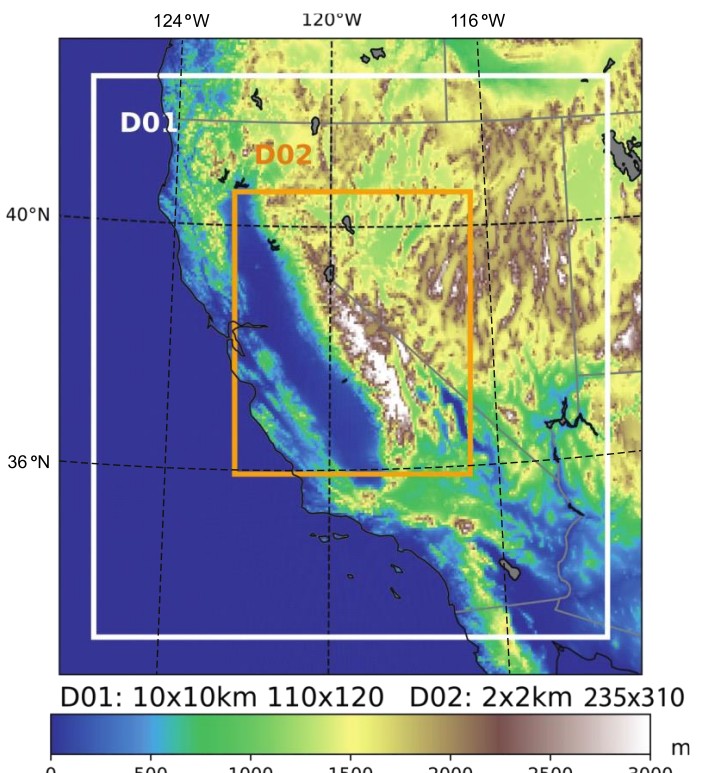

**Figure 1** WRF-Chem simulation domain 1 (D01) and domain 2 (D02) used in this study

**2.1.2 MERRA-2 and ERA5 reanalysis**

The Modern-Era Retrospective analysis for Research and Applications, Version 2 (MERRA-2) is

a widely used atmospheric reanalysis with a spatial resolution of 0.500°×0.625° and 72 vertical layers
(Buchard et al., 2017). MERRA-2 aerosol products are produced by combining GEOS atmospheric model
version 5 (GEOS-5) with a 3D variational data assimilation algorithm to incorporate satellite observations,
including Advanced Very High Resolution Radiometer (AVHRR), Moderate Resolution Imaging
Spectroradiometer (MODIS), and Multi-angle Imaging Spectro Radiometer (MISR), as well as ground-
based observations such as the AEronet RObotic NETwork (AERONET) (Gelaro et al., 2017). Although
the aerosol vertical profile, composition, and size distributions are not constrained by the assimilation of
aerosol optical depth (AOD), previous studies demonstrated that the aerosol assimilation system has
considerably improved the agreement with numerous observed aerosol properties (Buchard et al., 2016;



Buchard et al., 2017; Randles et al., 2017). The assimilation results in the imbalance of global dust mass
and produces a considerably larger deposition than the simulated dust emission (Buchard et al., 2017).
MERRA-2 simulates dust with diameter bins of 0.2–2.0 (DU001), 2.0–3.6 (DU002), 3.6–6.0 (DU003), 6.0–
12.0 (DU004), and 12.0–20.0 (DU005) μm, while the MOSAIC 4-bin in WRF-Chem simulates dust with
size bins of 0.039–0.156, 0.156–0.625, 0.625–2.5, and 2.5–10.0 μm. We therefore use the dust
concentrations of the first 4 size bins in MERRA-2 (DU001 + DU002 + DU003 + 0.74 * DU004) to match
with PM10 dust concentration in WRF-Chem (https://gmao.gsfc.nasa.gov/reanalysis/MERRA-
2/FAQ/#Q5).

ERA5 provides assimilated wind fields at a 0.25°×0.25° horizontal resolution at 137 hybrid

sigma/pressure levels from 1979 to near real time (Hersbach et al., 2020). This study obtained the 3-hourly
meridional and zonal wind field from February to June 2019 from 1000 to 500 hPa. The ERA5 wind
reanalyses were used with satellite-retrieved dust optical depth (DOD) to evaluate the classified dust
emission and transport patterns from the model.

**2.2 Satellite observations for validation**

The Infrared Atmospheric Sounding Interferometer (IASI) onboarded European Meteorological

Operation (MetOp) satellite series measures infrared radiation in 8,461 spectral channels between 3.63 and
15.5 μm. The instrument provides near-global coverage with a spatial resolution of 12 km at nadir (Hilton
et al., 2012) since 2007. IASI is primarily sensitive to coarse mode dust particles, and thus the retrieved
AOD at the wavelength of 10 μm can represent the DOD (Yu et al., 2019). Note that the thermal infrared
(IR) AOD reported by IASI is usually significantly smaller than the visible AOD in MODIS, because of
the spectral dependence of dust extinction (Zheng et al., 2022). We use the version 2.2 AOD product
developed at the Centre National de la Recherche Scientifique Laboratoire de Météorologie Dynamique
from https://iasi.aeris-data.fr/dust-aod/ (February 2022) (Capelle et al., 2014). The 0.3°×0.3° daily AOD
data covering California were produced by aggregating day and night retrievals at the satellite pixel

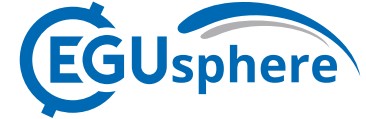

resolution (Capelle et al., 2018), in consideration of both data completeness and fine features. The 1.0°×1.0°
daily AOD was produced in a similar way to investigate dust transport from Asia across the North Pacific.

The MIDAS (ModIs Dust AeroSol) dataset provides global fine-resolution (0.1°×0.1°) daily DOD

between 2003 and 2017 using quality-filtered AOD from MODIS Aqua and DOD-to-AOD ratios from
MERRA-2 reanalyses (Gkikas et al., 2021). Despite the uncertainties in modeled DOD-to-AOD ratios, the
validations of the MIDAS dataset against the AERONET dust-like AOD and the LIdar climatology of
Vertical Aerosol Structure for space-based lidar simulation (LIVAS) DOD reveal a high level of agreement
at both global and station level (Gkikas et al., 2022). Compared with other MODIS-derived DOD products
(Song et al., 2021; Voss and Evan, 2020; Ginoux et al., 2012; Pu and Ginoux, 2018), MIDAS has finer
spatial and temporal resolutions over both land and ocean, which is particularly applicable in this study
focusing on a small region and a few cases at daily scale. The dataset has been extended to near real-time
to match our study year.

Cloud-Aerosol Lidar with Orthogonal Polarization (CALIOP) is a two-wavelength (532 and 1064

nm) polarization lidar onboarded the Cloud-Aerosol Lidar and Infrared Pathfinder Satellite Observation
(CALIPSO) satellite (Hunt et al., 2009). Since June 2006, the lidar has been collecting an almost continuous
record of high-resolution profiles of aerosol and clouds as fine as 30 m in the vertical, covering 82°N to
82°S (Winker et al., 2010; Winker et al., 2009). This study used clear-sky data from the CALIOP Version
4, level-2 aerosol profile product (Young et al., 2018) to investigate the vertical profile of elevated dust
layer, especially from remote transport. When there were large DOD shown in IASI and MIDAS, we
examined the vertical profiles of dust by identifying the "dust," "polluted dust," and "dusty marine" species
in the CALIOP data (Kim et al., 2018)

**2.3 SOM analysis**

We applied the self-organizing map (SOM), a clustering method developed in the field of artificial

neural networks, to recognize different weather features associated with dust transport and deposition.
Similar to other clustering methods, SOM projects high-dimensional data into a two-dimensional grid. SOM





has been widely used in atmospheric sciences to recognize spatially organized sets of patterns in the data
(Reusch et al., 2007; Bao and Wallace, 2015; Liu et al., 2022; Song et al., 2019). Before the machine-
learning process, the initiation nodes are assigned randomly or more efficiently, as used here, selected from
the leading empirical orthogonal functions (EOFs). During the training phase, the Euclidean distance
between each input pattern and the initiation nodes is calculated to begin an iterative procedure. The best-
matching node or the "winning" node is the one with the smallest distance between the initiation nodes and
the input vector. Then the winning node and the neighborhood nodes around the winner are updated to
adjust themselves toward the input vector. Since this process is iterated and fine-tuned, the nodes are self-
organizing. The final SOM nodes represent typical dust transport and deposition patterns across the Sierra
Nevada.

Here, we first used five variables from WRF-Chem in the SOM clustering, including dust

deposition flux at the Sierra Nevada, the low-level meridional and zonal dust transport fluxes, and the mid-
level meridional and zonal dust transport fluxes surrounding the Sierra Nevada. The 3 hourly model outputs
during February-June 2019 are used to count for the spatial distribution and temporal evolution of dust
transport and deposition. For WRF-Chem, we averaged the zonal and meridional dust fluxes in levels 3-5
(roughly 875-925 hPa over coastal California) to acquire the low-level transport features and averaged 200-
700 hPa fluxes to acquire the mid-level features. Levels 3-5 were selected to focus on airborne particulate
matter entrained above the planetary boundary layer and transported on the regional scale. Remote transport
of Asian and African dust is mostly found around 600–200 hPa, which flows downward to the lower
troposphere along the post-cold isentropic surface into the atmospheric river (AR) environment (Voss et
al., 2021). By selecting levels between 200-700 hPa, we were able to include all cross-Pacific remote
transport in the middle level. The choice of how many SOM nodes to prescribe is a trade-off between
distinctiveness and robustness. We found four-nodes clustering captures distinct transport patterns, while
more nodes produce redundant clusters with similar patterns.

To verify the recognized transport patterns based on WRF-Chem, we conducted SOM analyses

using variables from MERRA-2. We first remapped the same five variables using bilinear interpolation



from 0.5 ° × 0.625 ° to 10 km, the resolution of the WRF-Chem outer domain, before clustering. The vertical
levels of low-level and mid-level dust transport fluxes were selected to approximately match the WRF-
Chem pressure level. Four nodes were identified and arranged to make a direct comparison with those from
WRF-Chem. To further investigate if transport patterns recognized from SOM vary significantly with years,
we applied SOM analyses over 2011-2021 using MERRA-2 extended records of dust fluxes and deposition.

**3.   Results**
**3.1 Dust emission sources around the Sierra Nevada**

We find four emission source regions surrounding the Sierra Nevada where dust emissions could

potentially influence the mountain snow impurities between February and June (Fig. 2). The Mojave Desert,
located southeast of the Sierra Nevada, is characterized by low annual precipitation, sparse vegetation, and
dried fine soil. Airborne dust loading over the desert can reach 30000 ug m$^{-2}$ averaged over our study period
(Fig. 2a). It is generally transported eastward but can also be transported westward, influencing the southern
part of the mountain. Dust produced in the northern (Sacramento Valley) and the southern part (San Joaquin
and Tulare Basins) of the Central Valley is often transported eastward to the mountains. With high soil
aridity and a higher fraction of dry sand (Duniway et al., 2019), the southern Central Valley is more erodible
and emits a higher amount of fine dust. The Great Basin dust is relatively weak in magnitude but located at
a higher altitude. Therefore, it can easily ride along wind currents upward along the east slope of the
mountain. The column dust loading in MERRA-2 confirms our results in WRF-Chem (Fig. 2b), despite it
showing a stronger dust emission in the Great Basin while a weaker one in the Sacramento Valley. The
IASI shows the strongest IR DOD in the Mojave Desert, followed by the southern Central Valley, but
underestimates dust emissions from the Sacramento Valley (Fig. 2c). The underestimation is due to the fact
that IASI measures the radiation at IR wavelengths, which is more sensitive to coarse-mode dust particles
(Yu et al., 2019), whereas the fine dust produced in the Central Valley has a negligible contribution to DOD
at 10 μm. In contrast, MIDAS captures dust emissions from the Great Basin, the southern and northern
Central Valley (Fig. 2d) but not the Mojave Desert.  MIDAS is reported to underestimate DOD from the




Mojave Desert (compared to AERONET DOD) as MERRA-2 simulates lower dust amounts there (Gkikas
et al., 2021).

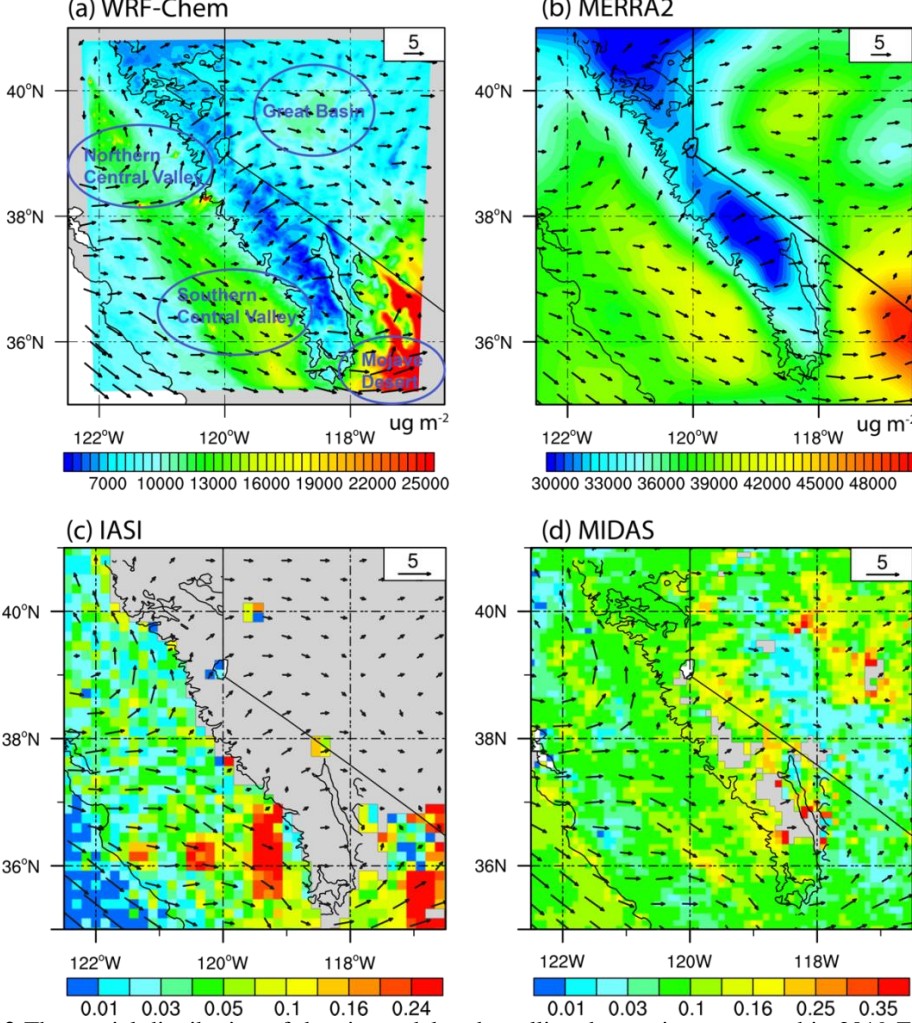

**Figure 2** The spatial distribution of dust in model and satellite observations averaged in 2019 February-
June. Column dust loading (ug m$^{-2}$) and low-level winds (roughly 875-925 hPa; m s$^{-1}$) in (a) WRF-Chem
and (b) MERRA-2. (c) Observed thermal infrared DOD at the wavelength of 10 μm from IASI (d) Observed
visible DOD at the wavelength of 550 μm from MIDAS. The low-level winds (m s$^{-1}$) in (c) and (d) are from
ERA5 reanalyses. Black contours indicate the elevation of 1500 m, which represents the Sierra Nevada
range used in this study. The grey area in c-d are missing pixels in satellite observations

**3.2 Dust transport across the Sierra Nevada**

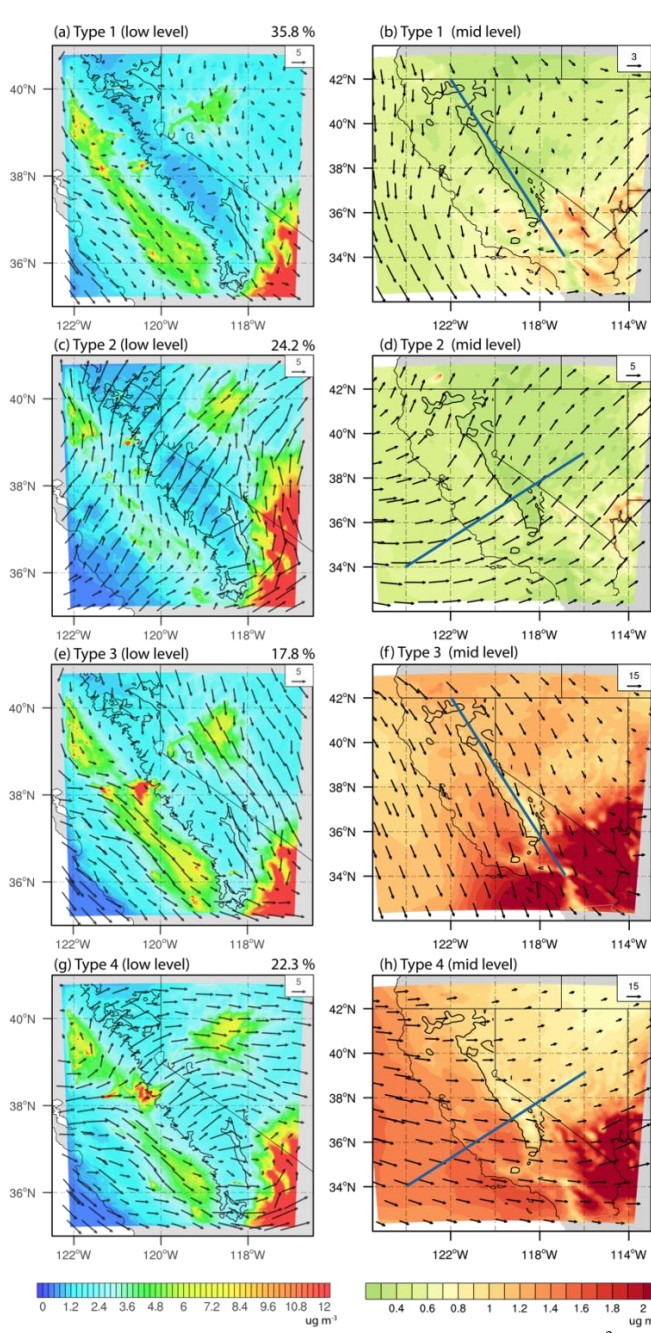

**Figure 3** (a, c, e, g) Low-level (roughly 875-925 hPa) dust concentration (ug m$^{-3}$) and wind vectors (m s$^{-1}$) in each of the four SOM type in WRF-Chem; The numbers on the top right of subplots denote the frequency of each type. (b, d, f, h) Mid-level (200-700 hPa average) dust concentration (ug m$^{-3}$) and dust transport in types 1-4; The position of the cross-section used for Figure 5 is denoted in each plot.



This section introduces the features of dust transport patterns discerned from WRF-Chem and
evaluates them against satellite observations over the period of February to June 2019. Figure 3 shows the
WRF-Chem dust concentration and wind in the low levels and middle levels averaged for each of the four
types acquired from the SOM analyses. The dust transport pattern represented in SOM type 1 accounts for
35.8% of hours from February to June (Fig. 3a), especially in February (43%) and March (57%) (Fig. 4a).
Type 2 occurs in 24.2% of the whole study period and contributes to more than 50% in February and then
decreases with the month. In contrast, types 3 and 4 account for 17.8% and 22.3%, respectively, with the
occurrence increasing with the month. The maximum occurrence is found in June for type 3 (40%) and
April for type 4 (34%), respectively.
**3.2.1 Mesoscale regional (MSR) transport**

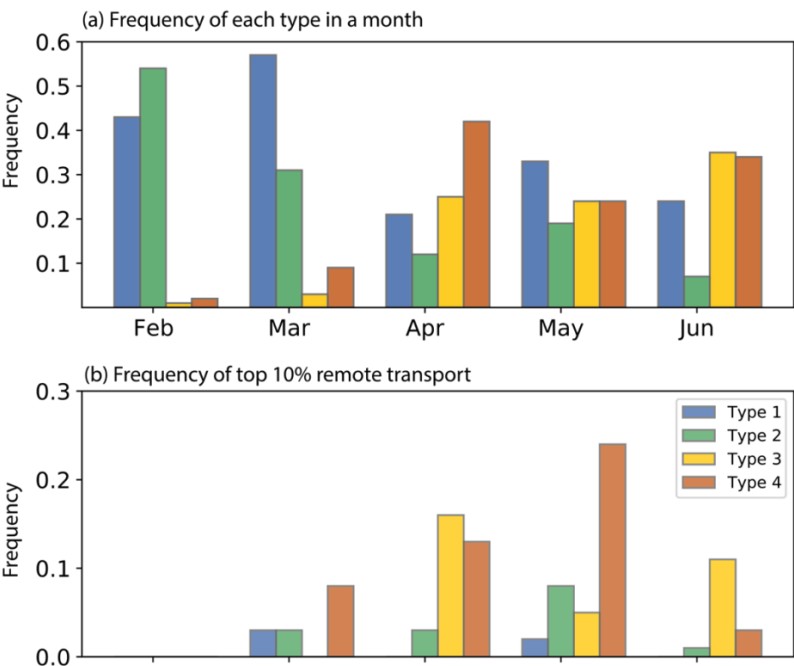

**Figure 4** (a) The frequency of each type (the time dominated by each type divided by total time in a month)
that occurs in February, March, April, May, and June in WRF-Chem. (b) The frequency of each type in the
top 10% remote transport (the time dominated by each type divided by total time of the top 10% remote
transport).

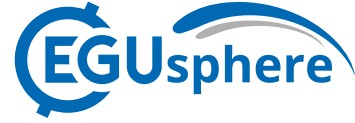

In type 1, dust is transported from northwest to southeast in the Central Valley in the low level
(roughly 875-925 hPa over the California coast). A vortex (Schultz Eddy) was found in the northern Central
Valley (Fig. 3a), circulating counter-clockwise and confining dust to the local environment (Bao et al.,
2008). The air inflow from the ocean is relatively weak and obstructed by the terrain. The Great Basin is
dominated by the northwesterlies. The emitted dust is transported southeastward and blocked by the
mountain, depositing dust on the east slope. Dust emitted from the Mojave Desert can be elevated to the
middle level (Fig. 3b). The cross-section further shows a vertical circulation where the Mojave Desert dust
is blown away from the Sierra Nevada at the low level and towards the mountain at 600-700 hPa (Fig. 6a).
A weaker mid-level cross-Pacific flow is found in type 1 than in other types (Fig. 5a), with no signals of
remote transport reaching the Sierra Nevada (Fig. 3b). Type 1 generally corresponds to the dust transport
in lack of prevailing large-scale weather systems. The high peaks of the Sierra Nevada produce mesoscale
circulations and prevent the Central Valley and Great Basin dust from being transported to the other side
of the mountain. It is referred to as the "mesoscale regional (MSR) transport" hereafter.

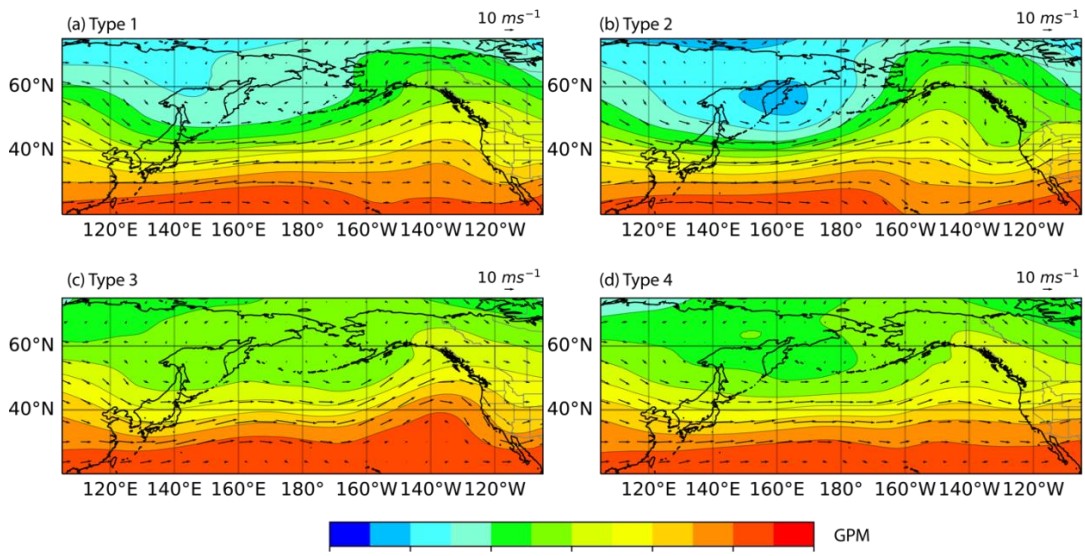

**Figure 5** Geopotential height (gpm) and wind vectors (m s$^{-1}$) at 500 hPa in each of the four SOM types in
WRF-Chem.

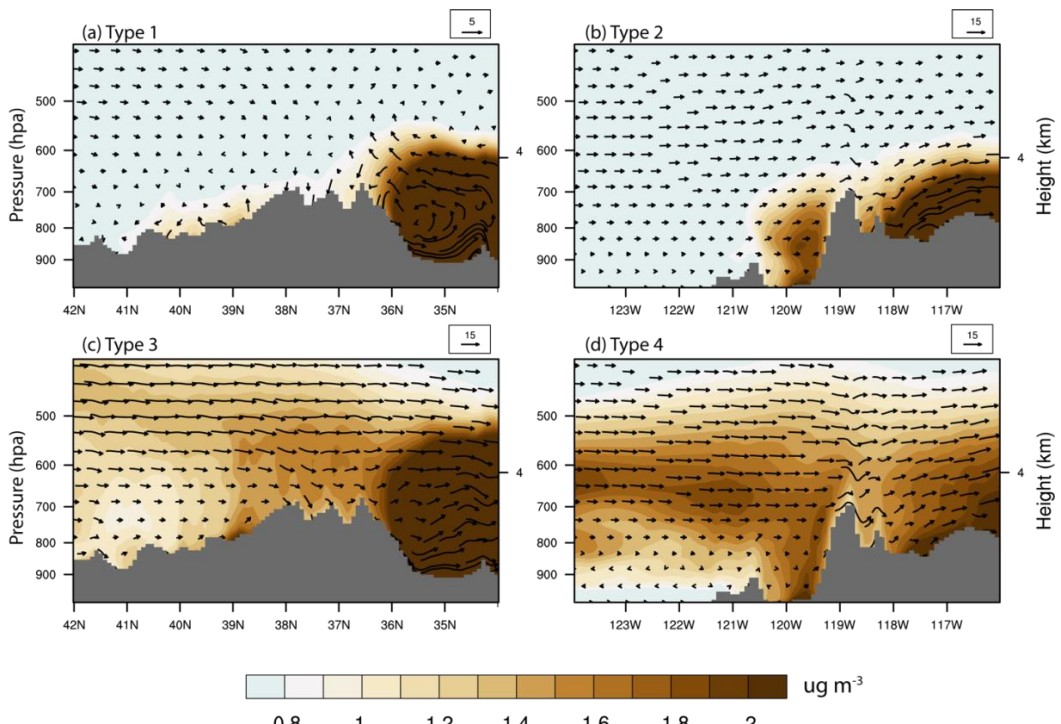

**Figure 6** Cross-section of dust concentration (shaded; ug m$^{-3}$) and dust transport fluxes (vectors; ug m$^{-2}$ s$^{-1}$) at 1000-400 hPa for each SOM type in WRF-Chem. The position of each cross-section is denoted in Fig. 3 b (Type 1), d (Type 2), f (Type 3), and h (Type 4). The grey area indicates the topography of the Sierra Nevada.

We validate the features of type 1 from WRF-Chem using satellite retrieved DOD and wind vectors from ERA5. The cloud contamination results in many missing satellite pixels in our study domain, making the transport patterns hard to discern on a single day. DOD and winds belonging to the same SOM type on consecutive days are averaged to maximize the data completeness. One typical example for each type is presented based on their representativeness and the maximum spatial coverage. Figures 7a-b present dust emission and transport patterns during May 10-12, a typical case for the MSR transport. In IASI, we find peak IR DOD (> 0.2) over the Mojave Desert and the southern Central Valley and moderate values in the Sacramento Basin related to the Schultz Eddy (Fig. 6a), resembling the relative magnitude of dust concentrations in regional source regions in WRF-Chem (Fig. 3a). MIDAS shows another evidence of dust transport pathways within the Central Valley with a higher resolution, although the maximum DOD shifts





slightly towards the mountain range (Fig. 7b). Dust emissions from the Great Basin are weaker than those
from the southern Central Valley.

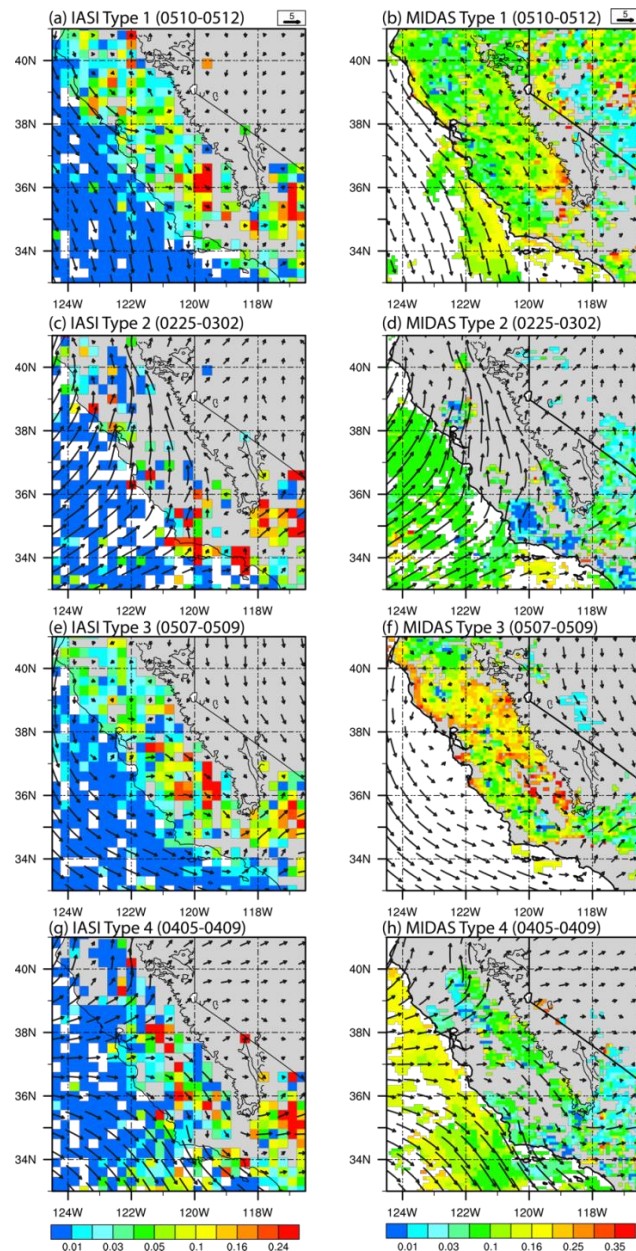

**Figure 7** (a,c,e,g) IR DOD at the wavelength of 10 μm retrieved from IASI and (b,d,f,h) visible DOD at
the wavelength of 550 μm from MIDAS for each type. The low-level winds (vectors; m s⁻¹) are obtained
from the ERA5 reanalyses. The numbers in the parenthesis indicate the event time period for the year 2019.





### 3.2.2 Sierra-barrier-jets-related (SBJ-related) transport


In type 2, the low-level winds turn to the north above the western slope of the Sierra Nevada (Fig.
3c), which resembles the terrain-locked Sierra barrier jets (SBJs) typically observed during the presence of
ARs (Neiman et al., 2013). The large-scale pattern consists of a low 500hPa geopotential height (GPH)
center in the north Pacific (Fig. 5b). The meridional gradient produces intense storm tracks from Kuroshio
Current towards Alaska. Indeed, we find extensive precipitations in type 2 (not shown), which produce
more wet deposition along the mountain's west slope and result in cleaner air in the Central Valley (Fig.
3c). The dust layer at the Central Valley is found below 700 hPa, mostly blocked by the high mountain
peaks and is hardly transported to the east slope of the mountain (Fig. 6b), despite the cross-barrier
westerlies found in the middle level. Dominated by SBJs, dust generated in the Great Basin and the Mojave
Desert is blown away from the mountain. No clear signal of remote transport is found on the California
coast (Fig. 3d). The dust transport from all sources is closely connected to SBJ; therefore, type 2 is referred
to as the "SBJ-related" transport. In both IASI and MIDAS, we find more missing pixels for SBJ-related
transport than any other type caused by cloud contamination (Figs. 7c-d). The AR-related landfalling
precipitations from February 25 to March 2 remove the airborne dust particles. A cleaner atmosphere might
be induced, but it is hard to confirm considering the missing pixels over the continent.

### 3.2.3 North-Pacific-High-related (NPH-related) transport


Type 3 has northwestern winds in both Central Valley and the Great Basin (Fig. 3e), transporting
Central Valley dust to the southwest part of the Sierra Nevada in early summer. It is known as the "North-
Pacific-High-related (NPH-related)" transport, during which the North Pacific High (NPH) built up in the
north Pacific 130° W produces the northwest-southeast wind direction along the California coast (Fig. 5c),
influencing the transport patterns for dust emitted from the surrounding sources. At the middle level, we
observe a meridional mid-level dust transport pathway (Fig. 3f), which appears at 400-500 hPa in the
northern Sierra Nevada and descends to 700 hPa at 36-37 °N, the top of the southern Sierra Nevada (Fig.
6c). As there are no major dust sources in the Pacific Northwest, the mid-level dust presumably origins



from Asia (discussed further in section 3.2.3). The dust emitted from the Great basin is transported by the
southward winds to the east slope of the mountains, while emissions from the Mojave Desert are transported
away from the mountain range.

The simulated dust concentration and transport in the NPH-related transport are confirmed by DOD

observations during May 7-9, with the transport pathway parallel to the California coast (Figs. 7e-f). Studies
have shown two main pathways of Asian dust transport to North America during the spring months: (1)
meridional excursions north into Alaska and then south along the U.S. west coast, and (2) zonal transport
over the North Pacific Ocean (Creamean et al., 2014). With north-south dust transport at the middle level,
the NPH-related transport characterizes the first pathway. To examine this hypothesis, we averaged the IR
DOD and 500 hPa wind field over the North Pacific during May 2-9. We included a few days before the
event (Fig. 8a) as it takes 7-10 days for dust to be transported from Asia to North America (Ault et al., 2011;
Creamean et al., 2013). The dust transport pathway shows that after being emitted from East Asia and the
Gobi Desert, dust is transported zonally to 150 °W, excursing north into Alaska/Canada and then traveling
south along the U.S. west coast. An elevated dust belt from 8 km to 12 km is discerned over the North
American coast (27 °N to 60 °N) from the CALIOP data, denoting the north-south transport of a thin dust
layer through the middle level (Fig. 8b).

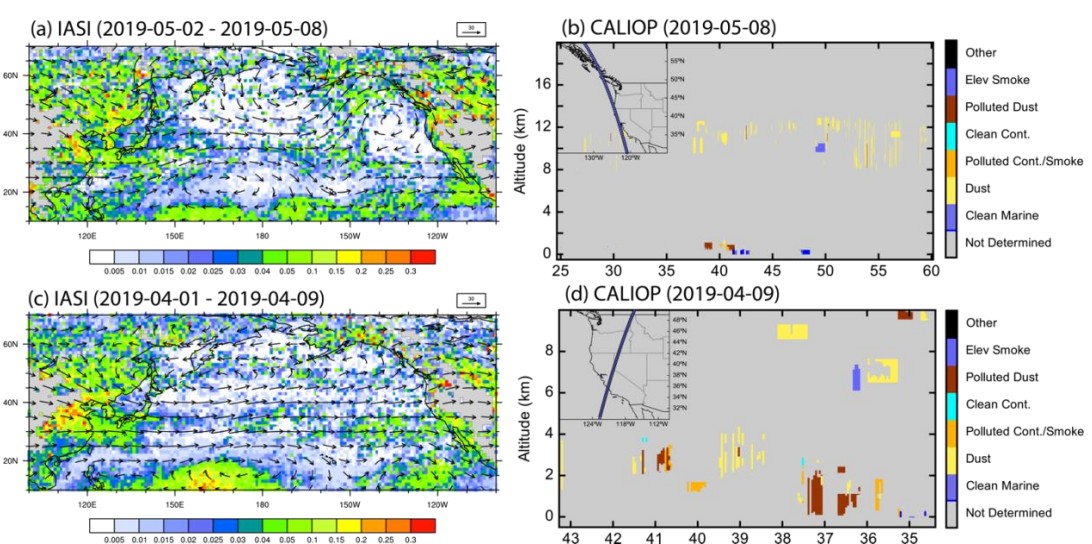




**Figure 8** (a) IR DOD from IASI and 500 hPa winds (m s⁻¹) from ERA5 over the North Pacific for a typical Type 3 case averaged between 2019-05-02 to 2019-05-08 (b) latitude-height cross-section of aerosol species from CALIOP on 2019-05-08 (Type 3); (c) same as (a) but for a typical Type 4 case averaged between 2019-04-01 to 2019-04-09; (d) same as (a) but for a typical Type 4 case on 2019-04-09

### 3.2.4 Cross-Pacific zonal (CPZ) transport

Air inflows from the ocean enter California and diverge to the northern and southern branches in type 4, transporting dust eastward across the Sierra Nevada (Fig. 3g). At the middle level, the low-GPH center recedes in April, and the isobars become straighter than in boreal winter, which facilitates the zonal transport of dust emitted from middle Asia over the North Pacific Ocean (Fig. 5d). The cross-section further shows an elevated dust layer is transported from the ocean at around 700-500 hPa (Fig. 6d). The concentrations are much stronger, and the altitude also lower than the NPH-related transport (Fig. 6c). The remotely transported dust descends to low altitudes when reaching the California coast and converges with the dust from the Central Valley at around 800 hPa. A portion of dust is compacted to the west slope at higher elevations, and the remaining across the mountains affects the east slope. Dust emitted from the Great Basin and the Mojave Desert is transported away from the mountains. Type 4 is denoted with "cross-Pacific zonal (CPZ) transport" to reflect the strong cross-Pacific dust transport.

April 5-9, a typical case for the CPZ transport, clearly shows the north and south branches of dust transport over the Central Valley (Figs. 7g-h). Different from the NPH-related transport pathway, the large-scale DOD and winds at 500 hPa (averaged over April 1-9) show that dust emitted from East Asia is being transported eastward, with a belt of IR DOD > 0.1 evident around 25-40 °N (Fig. 8c). The vertical distribution shows an elevated dust layer at 2-4 km above ground level, reaching the higher elevation of the mountain (Fig. 8d).

We calculated the mid-level dust remote transport, defined as the dust influxes from the north and west boundaries of the 200-700 hPa of WRF-Chem modeling domain 1, and investigated how the top 10% largest remote transport distribute in each SOM (Fig. 4b). Among all the large remote transport, CPZ transport accounts for 48% while NPH-related accounts for 32%, indicating that the zonal pathway plays a more important role in the cross-Pacific transport. Most remote transports are found in April and May, the



former dominated by the meridional transport in the existence of the NPH while the latter led by the CPZ
transport. The remaining two types contribute to a fairly small portion consistent with the clean atmosphere
in the middle levels (Figs. 3b, d).

To summarize, we discern four types of dust transport patterns across the Sierra Nevada and analyze

the monthly variability in their occurrence. The MSR transport represents the local dust transport, which
contributes to more than 20% of the time each month during February-June (Fig. 4a) in the absence of
prevailing weather systems. The SBJ-related air inflows transport dust eastward and are closely related to
the AR, during which the GPH and storm tracks at 500 hPa feature a typical large-scale pattern during the
boreal winter (Rodionov et al., 2007). As time evolves, the GPH center recedes, and the isobars become
more straight zonally in April, bringing dust from Asia and Africa to the western U.S. coast (CPZ transport).
In early summer, the buildup of NPH in the east Pacific corresponds to north-south winds along the
California coast, transporting dust along the Sierra Nevada (NPH-related transport).

**3.3 Dust deposition over the Sierra Nevada**

The averaged dust deposition and low-level dust transport for each type are shown in Fig. 9, including

both dry and wet depositions. The dry depositions consider the diffusion and gravitational effects, while
wet depositions describe in-cloud removal (rainout) and below-cloud removal (washout) by grid-resolved
stratiform precipitation as well as the sub-grid wet scavenging (Chapman et al., 2009; Easter et al., 2004).
In all SOM types, extensive depositions are found on the west slope in all types, generally decreasing with
elevation.

The MSR transport has the smallest deposition among the four types (Fig. 9a). Large depositions are

found in the southern Sierra Nevada and Lake Tahoe. Dust contributing to the deposition origins mainly
from the Mojave Desert and the Great Basin dryland. In contrast, large depositions found in the southern
and eastern parts of the mountains in NPH-related transport may be produced in agricultural land from the



southern Central Valley, as we find a persistent eastward transport pathway in the low level (Fig. 9c). The
remote transported dust plays a minor role as it is located above 8 km in altitude.

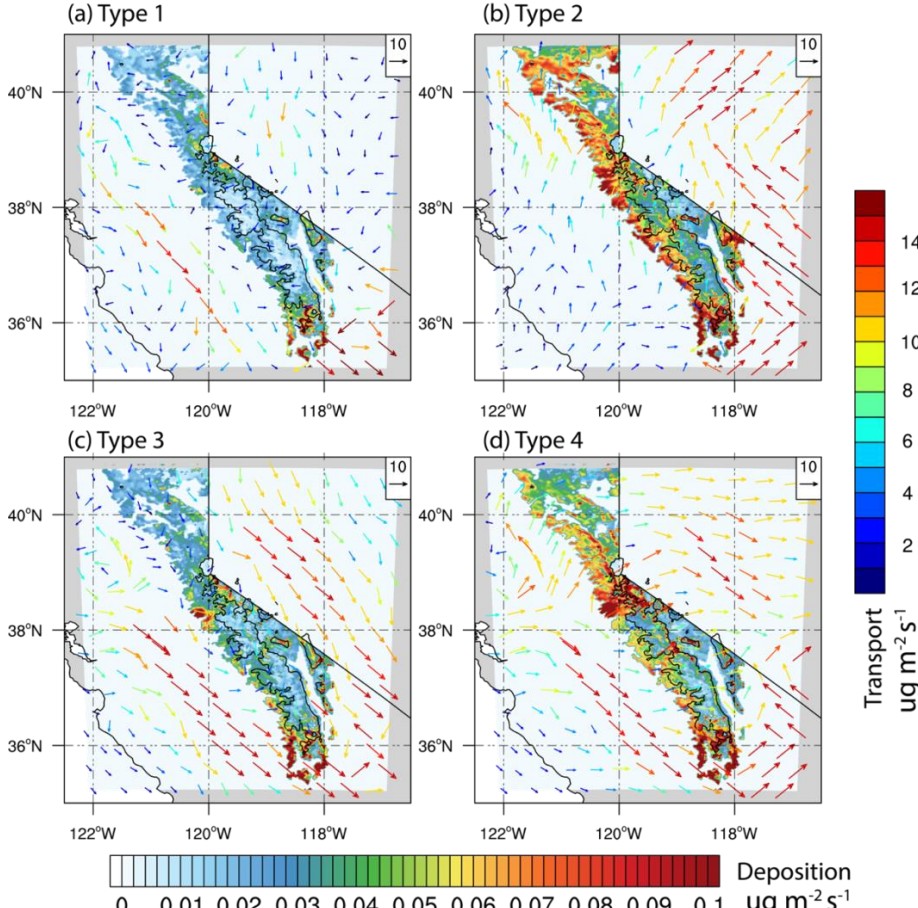

**Figure 9** (a-d) Dust deposition (shaded; ug m$^{-2}$ s$^{-1}$) over the Sierra Nevada and low-level dust transport
fluxes (colored vectors; ug m$^{-2}$ s$^{-1}$) across the Sierra Nevada averaged over each of the four SOM types in
WRF-Chem. Black contours indicate an elevation of 2500 m. The bottom color bar shows the magnitude
of dust deposition over the Sierra Nevada while the right color bar shows the magnitude of dust transport
flux vectors.

While SBJ-related transport has the lowest low-level dust concentration over the Central Valley, it

produces the largest deposition along the west slope (Fig. 9b). Most eastward transport in the southern
Sierra Nevada is obstructed by the high mountain peaks, resulting in large depositions below 2900 m. The
SBJ turns eastward in the Sacramento Basins and climbs through the mountain north of 38 °N, producing



a relatively homogenous deposition in the northern part. The combination of dust transport and deposition
indicates that dust influencing the mountain snow impurities mostly comes from the Central Valley.
Compared with the other SOM types, SBJ-related transport has large depositions at elevations higher than
2500 m (discuss later). Large depositions are also found in the CPZ transport (Fig. 9d), with the largest
value occurring on the west slope of the central and southern Sierra Nevada, contributed by both Asian dust
and Central Valley dust. Compared to the MSR and NPH-related transport, the large-scale westerlies in the
Central Valley (SBJ-related and cross-Pacific transport) produce larger deposition, probably because of the
more efficient removal of particles by collision with terrestrial surfaces at higher elevations (Fig. 6d).

To quantify the relative importance of wet and dry depositions in each 3 hourly total deposition data,

we calculate the fraction of wet depositions to total depositions averaged over the Sierra Nevada:
$\frac{Wet\ deposition}{Wet\ deposition+Dry\ deposition}$. The contribution of dry deposition is defined in a similar way. We find the
wet deposition accounting for 40% in frequency in the SBJ-related type. The landfalling precipitation has
deposited large amounts of airborne dust on the snow surface, producing a cleaner atmosphere as we have
found in Fig. 3c. The frequent wet depositions also explain the larger depositions in high elevations (Fig.
9b): dust particles reaching the high mountains are small in size and difficult to deposit through gravitational
effects. Wet deposition is a more efficient way of depositing small particles as they collect dust in raindrops.
In contrast, the dry depositions play predominant roles (more than 80% in frequency) in all the other types
(Fig. 10a). Figure 10b further shows the contribution of wet deposition increases with deposition intensity.
The averaged contribution of wet depositions in magnitude increases from 19% in all events to 29% in the
top 10 percentile, 36% in the top 5 percentile, and 56% in the top 1 percentile largest events, supporting
our conclusion that wet deposition is a more efficient way of dust deposition.





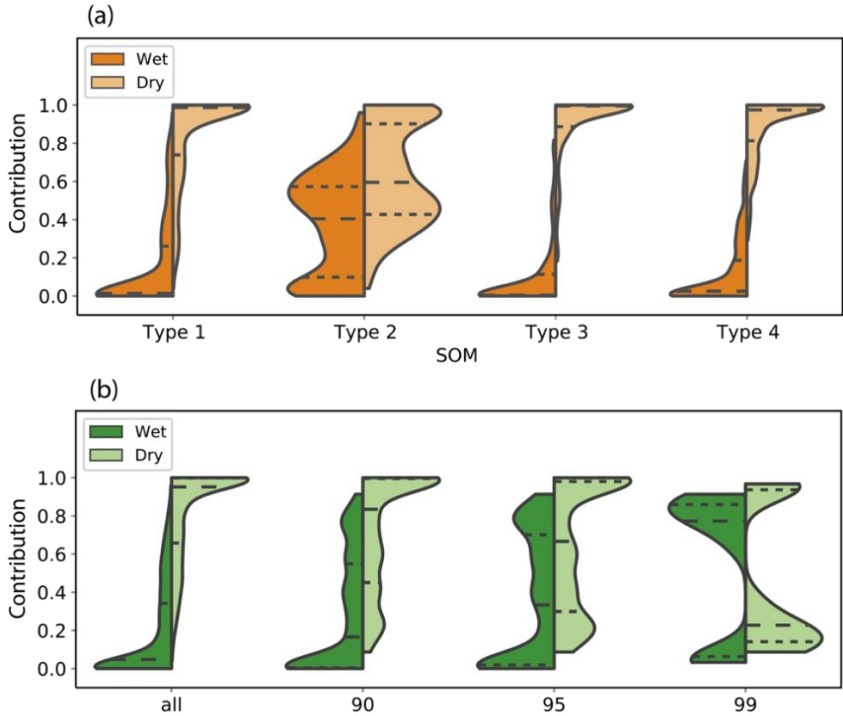

**Figure 10** (a) Distribution of contribution of wet and dry depositions to total deposition in each type in WRF-Chem. (b) Distribution of contribution of wet and dry depositions to total deposition for all deposition, depositions over $90^{th}$, $95^{th}$, and $99^{th}$ percentile. The three lines inside violin plot (a-b) indicate 25%, 50%, 75% of the distribution

**3.4 Features of the dust transport in MERRA-2**

We repeated the SOM analyses using 2019 MERRA-2 data to examine the WRF-Chem model performance and interannual variability. We conducted additional SOM analyses using 2011-2021 climatology MERRA-2 data to investigate the interannual variability of the transport patterns. The low-level and mid-level dust transport features identified in MERRA-2 (Figs. 11-12) are similar to their corresponding types in WRF-Chem (Fig. 3), with types 1, 2, 3, and 4 representing MSR, SBJ-related, NPH-related and CPZ transport, respectively (Fig. 11). Additionally, north-south transport occurs in the middle layer in type 3 and west-east transport in type 4, despite the slight difference in the peak region (Fig. 12).



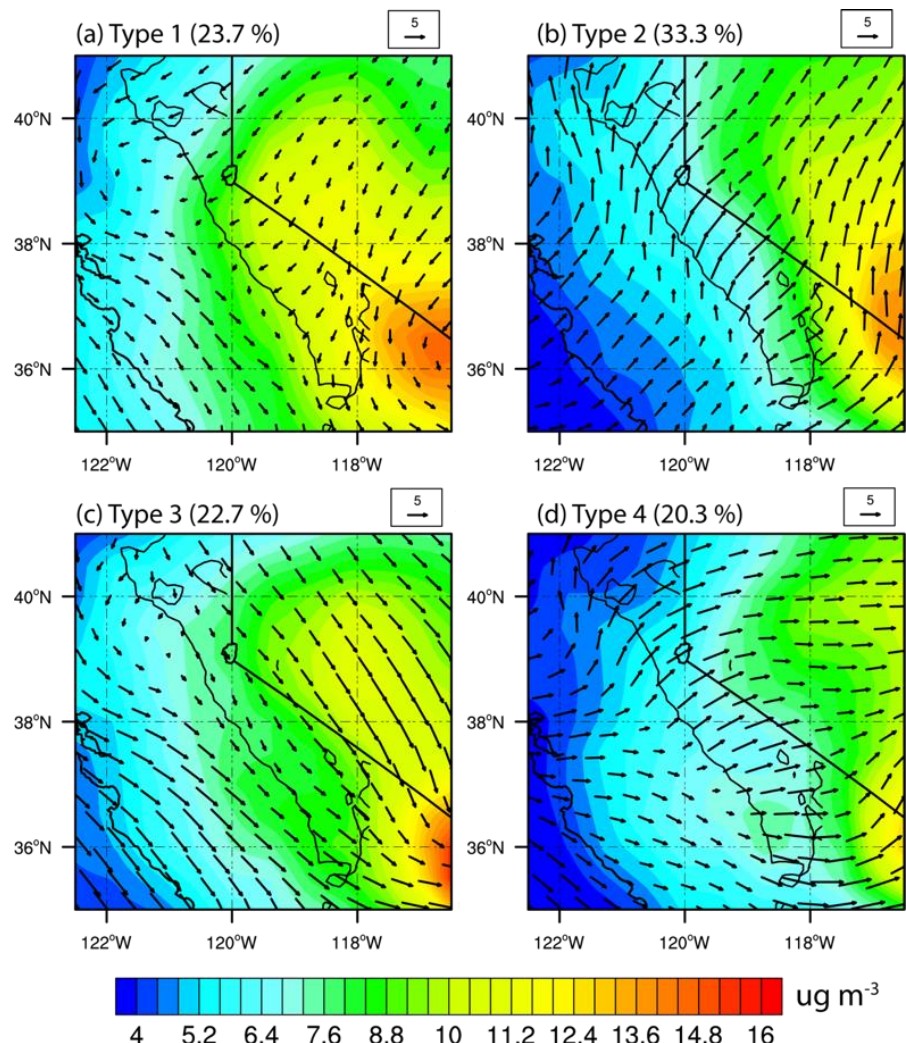

**Figure 11** Low-level dust concentration (ug m⁻³) and wind vectors (m s⁻¹) in each of the four SOM types from MERRA-2 for the year 2019. The numbers on the top of subplots denote the frequency of each type.

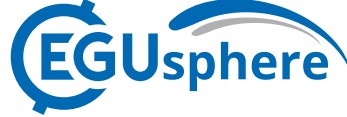

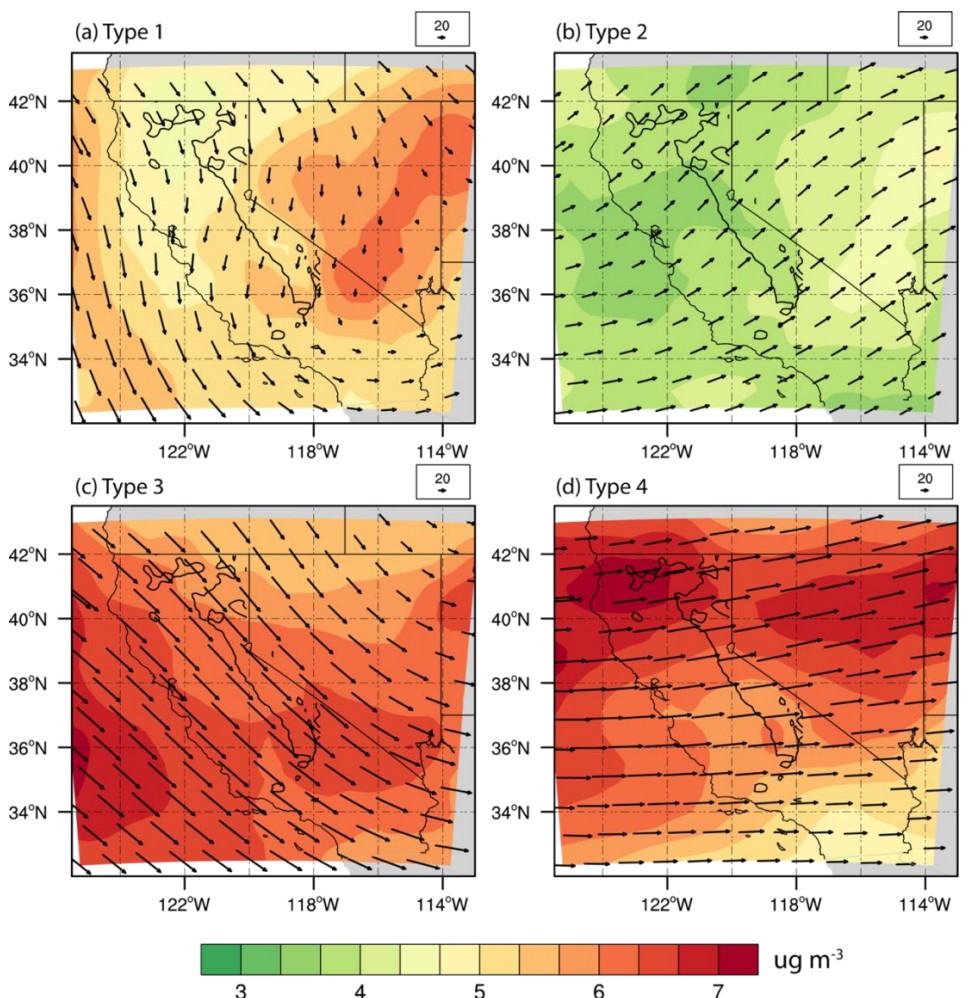

**Figure 12** Mid-level (200-700 hPa average) dust concentration (ug m$^{-3}$) and dust transport fluxes (ug m$^{-2}$
s$^{-1}$) in each of the four SOM types from MERRA-2 for the year 2019

The relative contribution of each transport type in MERRA-2 (SBJ-related > MSR > CPZ > NPH-
related) is generally consistent with the results in WRF-Chem (MSR > SBJ-related > CPZ > NPH-related),
except that the MSR transport occurs less frequently in MERRA-2. The difference is largely caused by the
spatial resolution of the two datasets. With a resolution of 0.5°×0.625°, MERRA-2 has smooth topography
information and cannot resolve the high peaks of the Sierra Nevada which produce the MSR winds and
transport. Consequently, MSR transport contributes to a smaller fraction in the MERRA-2. The coarser

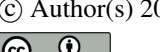

resolution MERRA-2 also produces a more homogeneous dust concentration at low levels than 2-km WRF-
Chem.
Similar dust concentrations and transport patterns are found in the 11-year SOM analysis (Fig. 13),
indicating that the four patterns identified in 2019 are representative of the climatological conditions. In
climatology, the SBJ is weaker and air inflows hit the California coast at a further north latitude (about
40 °N; Fig. 13b), which is reasonable as 2019 is a typical El Nino year with stronger AR reaching California
further south than usual. The changes in the transport patterns reflect the interannual variations of large-
scale forcings and regional weather conditions.

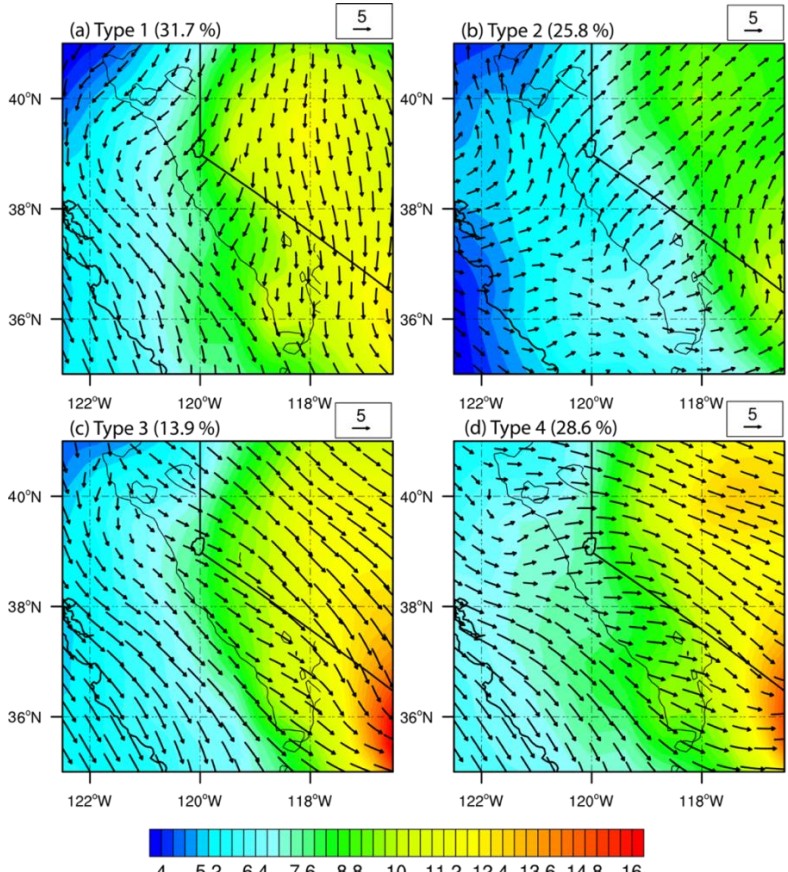

**Figure 13** Low level dust concentration (ug m$^{-3}$) and wind vectors (m s$^{-1}$) in each of the four SOM types
from MERRA-2 averaged over 2011-2021. The numbers on the top right of subplots denote the frequency
of each type.





**4. Conclusions and discussion**

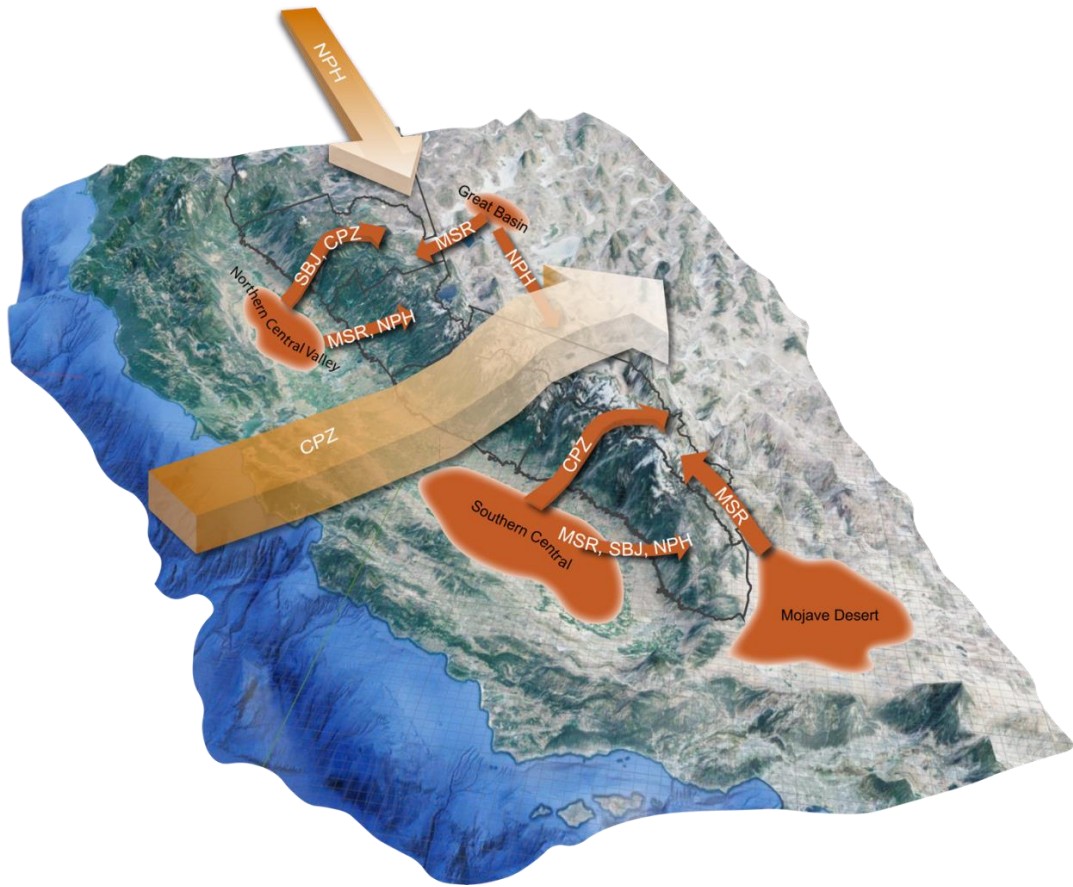

**Figure 14** Schematic diagram of typical dust transport patterns across the Sierra Nevada. The "MSR"
demotes mesoscale regional transport. The "SBJ" and "NPH" denotes dust transport dominated by Sierra-
Block Jets (SBJ) and North Pacific High (NPH), respectively, while the "CPZ" denotes Cross-Pacific Zonal
transport.

With a focus on the dust that influences the mountain snow, we investigated the dust sources

surrounding the Sierra Nevada and their typical transport patterns during the spring and early summer.
Despite the strongest emissions from the Mojave Desert, dust is only transported northward to the mountain
when the mesoscale weather pattern dominates the southwest U.S. (Fig. 14). During 64.25% of our study
period, dust from the Mojave Desert is transported away from the mountains. Dust emitted from the Great
Basin is transported to the central Sierra Nevada during MSR transport and to the eastern part when the



NPH builds in the eastern Pacific. It is blown eastward by air inflows from the ocean during SBJ or cross-
Pacific transport. In contrast, dust produced by the Central Valley is persistently transported to the west
mountain slope, playing an essential role in snow impurities there. Carried by intense air inflows, it can be
transported to the eastern slope of the Sierra Nevada.

During April, Asia dust is transported zonally over the North Pacific through the straight zonal

isobars at the middle level. The dust layer descends to 800 hPa when it reaches the California coast. In the
presence of the NPH, dust emitted from Asia excurses north into Alaska/Canada and travels south along
the U.S. west coast. The dust travels at a higher altitude, and the concentrations are weaker than the zonal
transport.

Large amounts of depositions are found on the west slope, which generally decrease with elevations.

Dust particles transported to the higher altitude are small in size and difficult to deposit through gravitational
effects. The SBJ-produced AR collects dust in the rain and snow and deposits it on the high mountain.
Besides, considerable depositions occur when the elevated dust layer from the Pacific collides with the
mountain.

We acknowledge that our characterization of dominant transport patterns might be limited by model

uncertainties. Besides, the coarse-resolution reanalyses data, MERRA-2, cannot accurately resolve the
topography effects and tends to underestimate mesoscale regional transport. Furthermore, both WRF-Chem
and MERRA-2 describe dust emissions from dryland by relating them to high wind speed, soil moisture,
and soil type (Ginoux et al., 2001), while dust emission from agricultural lands is not specifically
implemented. However, a comprehensive evaluation of airborne dust and PM2.5 concentration between
model simulation and site observations in our previous study shows a good agreement between both (Huang
et al., 2022a). In addition, the dust transport pathways have well-defined patterns associated with the
mesoscale and large-scale weather systems. The general consistency across different models (WRF-Chem
and MERRA-2) and observations (satellite analysis) and across different years give us confidence that the
results are valid despite model uncertainties.



The analyses of dust emission and transport can be used to understand dust transport in a changing
climate. Studies have shown that global warming continues to dry the soil, producing more dust emissions
over the western U.S. Nevertheless, the change in transport and deposition patterns has not been well
recognized. Our study highlighted the connection between dust transport and dominant weather patterns
across the Sierra Nevada; the latter might respond in a more predictable way to climate change. Future
projections show that global warming may increase the frequency of landfalling AR by 20-35% by the end
of the 21$^{st}$ century (Hagos et al., 2016; Rhoades et al., 2021). Besides, the widening of the Hadley Cell in
response to global warming might enhance the NPH and shift it poleward (Song et al., 2018; Choi et al.,
2016). Thus said, the SBJ- and NPH-related dust transport may occur more frequently while the MSR
transport may become less common. In this regard, changes in dust emissions from the Central Valley might
play a more critical role in mountain snow impurities than those from the Mojave Desert and the Great
Basin, producing more depositions on the west slope of the Sierra Nevada.

**Data availability:**
The IASI DOD data is acquired from https://iasi.aeris-data.fr/dust-aod/. The MIDAS DOD is acquired from
https://zenodo.org/record/4244106#.YsJqe-zMIws. MERRA-2 aerosol reanalyses are available from
https://disc.gsfc.nasa.gov/datasets?keywords=MERRA2&page=1 and ERA5 wind reanalyses are available
from https://rda.ucar.edu/datasets/ds633.0/. The WRF-Chem and MERRA-2 SOM clustering results have
been uploaded to https://doi.org/10.5281/zenodo.6795994.

**Author contributions:**
HH performed the analysis and drafted the manuscript. The methodology was developed by HH and YL.
JZ and AG provided the observational data used for model validation. YQ, CH, and ZZ helped with the
analysis and offered valuable comments. All authors contributed to writing and editing the manuscript.

**Competing interests:**

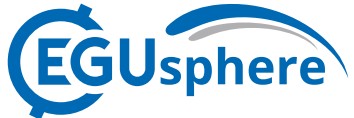

The authors declare that they have no conflict of interest.

**Acknowledgement:**
This research was supported by NASA awards: 80NSSC21K0997, 80NSSC20K1722, 80NSSC20K1349,
and 80NSSC18K1489. Antonis Gkikas was supported by the Hellenic Foundation for Research and
Innovation (H.F.R.I.) under the "2nd Call for H.F.R.I. Research Projects to support Post-Doctoral
Researchers" (project acronym: ATLANTAS, project number: 544). The Pacific Northwest National
Laboratory (PNNL) is operated for DOE by the Battelle Memorial Institute under contract DE-AC05-
76RLO1830.



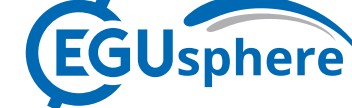

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
