# Peer review of "Where does the dust deposited over the Sierra Nevada snow come from?"

_EGUsphere, 2022_

## Author Comment (AC1)

**Reviewer #1**

**Review of the paper "Where does the dust deposited over the Sierra Nevada snow come from?" by Huang et al., submitted for publication in ACP.**

**General comments**

The paper aim at characterising the circulation types associated with dust deposition over the Sierra Nevada snow cover, by using SOM based clustering. The paper is well written, in a clear and concise manner. Motivations and objectives are clearly declared and put in the context of the existing knowledge. Methods are appropriated and clearly described. Conclusions follow from evidence. The findings discussed in the paper are relevant, and worth of publication. However, I believe that the paper could be further improved with a few modifications.

We thank the reviewer for providing positive and constructive comments to improve our analyses. We have made more clarifications and discussions based on the reviewer's comments. Below we provide a point-by-point response to the comments. The reviewer's comments are in black font in the following paragraphs, and our responses are in blue.

My first and main suggestion is motivated by the title of the paper, which questions the origin of dust deposition over the Sierra Nevada. I wonder why the authors didn't corroborate their findings by adding a back trajectories analysis. I really believe that the robustness of the results would significantly benefit from such an analysis, especially the part concerning the long-range transport.

R: We appreciate the reviewer for the suggestions on validating our SOM results with back trajectory analyses. In the revised version, we have conducted back trajectory analyses using Hybrid Single-Particle Lagrangian Integrated Trajectory (HYSPLIT) model from NOAA with Meteorological forcing from the North American Mesoscale Forecast System (https://www.ready.noaa.gov/hypub-bin/trajtype.pl?runtype=archive). We selected typical days for the four SOM types as in Figure 7 and we specifically chose three sites in Central Sierra Nevada (38°N, 120.3°W), Southern Sierra Nevada (36.5°N, 119°W), and Eastern Sierra Nevada (37°N, 117 °W), to represent dust deposition at different subregions. Based on the back trajectory analysis, two new figures (Figure R1 and Figure R2) were added to the supplementary material (Figs. S2-3), and the following discussion has been added to the revised manuscript (Lines 407 – 424 in the track change version).

"We further conducted air mass back trajectory (AMBT) simulations to evaluate the dust emission sources and transport pathways identified using SOM analyses. The back trajectory simulation was conducted using Hybrid Single-Particle Lagrangian Integrated Trajectory (HYSPLIT) model with meteorological forcings from North American Mesoscale Forecast System. We selected typical days for the four SOM types as in Figure 7 and three sites located at the Central Sierra Nevada (38 °N, 120.3°W), Southern Sierra Nevada (36.5 °N, 119 °W), and Eastern Sierra Nevada (37 °N, 117 °W), to represent dust deposition at different subregions.

The results of 12-hour and 7-day AMBT results corroborate the identified local and long-range transport pathways for each type. The transport pathways generally follow the wind directions shown in Figure 7. Multiple emission source regions are found in type 1, including the Central Valley where dust is transported eastward to the windward slopes and the Great Basin where dust is transported westward to the lee-side slopes (Fig. S2). In type 2, dust deposited in all three sites comes from the Central Valley (Fig. S2), and the transport corresponds to the direction of SBJ during AR (Fig. 7c). Types 3 and 4 are affected by both local and remote transport. Locally, Dust mainly comes from the northern California and the Great Basin in type 3, while it comes from the Central Valley in type 4. Remotely, in type 3, we find dust emitted from Asia and North Africa excurses meridionally to Alaska at 135° W and then travels southward along the U.S. West Coast (Fig. S3a). In contrast, dust emitted from east Asia is transported zonally across the Pacific, reaching the Sierra Nevada from the west (Fig. S3b)."

[Figure]

Figure R1 12-hour backward trajectories of the air masses reaching the three sites in the central (left column), south (middle column) and eastern (right column) Sierra Nevada on

typical days of four SOM types. The air mass back trajectory data was obtained using the hybrid single-particle Lagrangian integrated trajectory (HYSPLIT) model from the National Oceanic and Atmospheric Administration (NOAA). Air mass back trajectories were calculated at 1000 m above ground level.

[Figure]

Figure R2 7-day backward trajectories of the air masses reaching the east Sierra Nevada site on typical days of type 3 (a) and type 4 (b). Air mass back trajectories were calculated at 3000 m above ground level.

I also wonder why the authors only analysed one year. Possibly because of limited computational resources. However, reliable reanalysis products are available (MERRA, CAMS) for analysis on the long term (even longer than the 11-year climatology presented in the paper), which could then be used to select one or more interesting years for more WRF simulations.

R: Per the reviewer's suggestions, we have extended the MERRA-2 analyses from 2011-2021 to 2001-2021 and added more discussions regarding the interannual variability and extreme years. We do not include data before the year 2000 as it does not use MODIS AOD in the assimilation and would bring more uncertainties due to the difference between assimilation data (with MODIS AOD after the year 2000 and without MODIS before 2000). The updated SOM clustering results show consistent patterns in low level and the mid-level dust transport as our original results using data from 2011-2021 (Fig. R3). Type 1, the MSR (mesoscale regional transport) has the largest frequency, followed by type 2 (SBJ-related transport), type 3 (NPH-related transport), and type 4 (cross-pacific zonal transport).

[Figure]

Figure R3 a) Low-level dust concentration (ug m$^{-3}$) and wind (m s$^{-1}$) and b) mid-level (200-700 hPa average) dust concentration (ug m$^{-3}$) and dust transport fluxes (ug m$^{-2}$ s$^{-1}$) in each of the four SOM types from MERRA-2 for the year 2001-2021. The numbers on the top right of subplots denote the frequency of each type.

We then computed the frequency of each type in a year during 2001-2021 (Fig. R4). Interestingly, we find types 1 and 4 have an opposite correlation coefficient (R=-0.75) in their frequency, indicating the competing impact between remote transport and local emissions on dust over the Sierra Nevada. Especially, type 4 tends to occur more frequently during La Niña years (for example, 2008, 2011, and 2021 springs, denoted by red bars) while less frequently during El Niño years (2015, 2016, and 2019 springs, denoted by blue bars). An opposite conclusion can be drawn for type 1. Therefore, we further examine the SOM pattern and frequencies during the three La Niña years and three El Niño years.

We found similar dust transport patterns and relative contribution of four types during the ENSO years compared to the climatology (Figs. R5-R6). However, we note that La Niña years (Fig. R5) have stronger low-level and mid-level dust concentrations than El Niño years (Fig. R6). During La Niña years, the frequencies of types 3 and 4 are higher than during El Niño years, reflecting the increased contribution of cross-Pacific remote transport to the dust concentration over California. The increase in remote transport weakens the relative contribution of local sources, decreasing the frequency of type 1.

We agree that the analyses of extreme years from MERRA-2 can be used to select more interesting years for WRF simulations, which can be extended in our future studies. We update the corresponding figures (Fig. S4-S8) and add the above discussions in Lines 504 to 516.

[Figure]

Figure R4 The frequency of each type during 2001-2021. The red bars denote La Nina years while the blue bars denote El Nino years.

"The changes in the transport patterns reflect the interannual variations of large-scale forcings and regional weather conditions, which is investigated using the frequency of each type in a year during 2001-2021 (Fig. S4). Types 1 and 4 have a negative correlation coefficient (R=-0.75) in their frequency, indicating the competing impact between remote transport and local emissions on dust concentrations over the Sierra Nevada. Especially, type 4 tends to occur more frequently during La Niña years while less frequently during El Niño years. An opposite conclusion can be drawn for type 1.

We further examine the dust transport pattern and the frequency of the four SOM types during three La Niña (2008, 2011, and 2021) and three El Niño (2015, 2016, and 2019) years. We find that the La Niña years have larger dust concentrations than El Niño years in both lower levels and middle levels (Figs. S5-S8), due to suppressed precipitations and drier soil in the southwestern U.S. Meanwhile, the frequencies of types 3 and 4 are higher in El Niño years, reflecting the increased contribution of cross-Pacific transport to dust loading over California. The increase of remote transport weakens the relative importance of local emissions, decreasing the frequency of type 1."

[Figure]

Fig. R5 a) Low-level dust concentration (ug m⁻³) and wind (m s⁻¹) and b) mid-level dust concentration (ug m⁻³) and dust transport fluxes (ug m⁻² s⁻¹) in each SOM type averaged over La Niña years (2008, 2011, 2021). The numbers on the top right of subplots denote the frequency of each type.

[Figure]

Fig. R6 The same as Figure R5, but for El Niño years.

**Specific comments**

L6: what is "the dust prone area"? Can you be more specific?

R: The dust-prone area refers to the dust emission regions, especially global drylands that are prone to degradation and desertification. We have replaced the world with "global drylands".

"Dust emission is an integral part of aridification and mirrors the effects of climatic change and anthropogenic land use on global drylands (Duniway et al., 2019)."

L39: this sentence is not clear, premature mortality is rather a consequence of air-quality associated diseases (which include cardiovascular illnesses) than a direct impact. Please rephrase to highlight direct and indirect impacts of dust on health.

R: We agree. The sentence has been revised as follows (Lines 39-42):

"Research has indicated that exposure to dust particles can cause respiratory infections, heart disease, and chronic obstructive pulmonary disease (COPD) (Laden et al., 2006; Lim et al., 2012; Crooks et al., 2016). A significant association between dust exposure and increased mortality has been reported, but there is no consensus in this regard to date (Giannadaki et al., 2014)."

L183: too generic, please clarify what you mean by "projecting high-dimensional data into a two-dimensional grid".

R: As shown in Fig. R7, each gray square represents high-dimensional data (in this study it represents a combination of 2D variables including dust deposition, low-level dust transport fluxes, and mid-level dust transport fluxes). The two-dimensional grid is denoted by the black dot which represents the typical features of a cluster of high-dimensional data. We agree that the description might be too generic and confusing and have removed the sentence.

[Figure]

Fig. R7 Concept Diagram of SOM

L198: are these model levels?

R: Yes. The levels 3-5 are WRF-Chem lowest levels 3-5. Because WRF uses a hybrid terrain-following vertical coordinate, levels 3-5 correspond to different pressures over coastal California (approximately 900-950 hPa) and the Sierra Nevada (approximately 700-650 hPa). We revised the sentence as follows (Lines 200 - 202):

"For WRF-Chem, we averaged the zonal and meridional dust fluxes in model levels 3-5 (roughly 900-950 hPa over coastal California and 650-700 hPa over the Sierra Nevada) to acquire the low-level transport features"

L206: did you select the number of modes subjectively by testing the method, or did you use any objective methodology to assess distinctiveness and robustness? The approach used should be clarified.

[Figure]

Fig. R8 Classifiable index as a function of the number of clusters.

R: We used the classifiable index (CI) to evaluate the robustness of the clustering process and determine the number of clusters (Vigaud and Robertson, 2017; Vigaud et al., 2018; Hannachi, 2010). We tested the number of nodes/clusters (k) that ranges in 3, 4, 5, 6, 8, 9, and 16. For each $k$, the CI was constructed using the minimum spatial correlation coefficient between the clusters obtained from the full data and many random halves (100 halves used here) of the data (Hannachi, 2010). Therefore, the CI measured the reproductivity of the $k$ clusters partitioning (Visbeck et al., 2001), with perfect partitioning leading to 1. Figure R8 shows the CI as a function of the number of clusters for the year 2019 using WRF-Chem output. With the highest CI, the 4-cluster partition is shown to well represent distinct dust transport and deposition patterns over the Sierra Nevada and is then used in our analysis. The discussion of the CI test is added in Lines 208-216 and the figure is added to Fig. S1

"We tested the number of clusters ($k$) that ranges in 3, 4, 5, 6, 8, 9, and 16 to assess the distinctiveness and robustness of different $k$. For each $k$, the robustness of the clusters was measured by a classifiability index (CI) (Vigaud and Robertson, 2017; Vigaud et al.,

2018; Hannachi, 2010) constructed using the minimum spatial correlation coefficient between the clusters obtained from the full data and many random halves of the data (100 halves used here) (Hannachi, 2010). Therefore, the CI measured the reproductivity of the $k$ clusters partitioning (Visbeck et al., 2001), with perfect partitioning leading to 1. Figure S1 shows the CI as a function of the number of clusters using WRF-Chem output for 2019. With the highest CI, the 4-cluster partitioning well represents distinct dust transport and deposition patterns over the Sierra Nevada and is used in this study."

[Figure]

Figure R8 Classifiable index as a function of the number of clusters.

Also, how many EOFs have been retained before clustering? On which spatial domain the SOM method has been applied?

If I understand it correctly, the reviewer asks if we have conducted any filtering to reduce the dimension of the input vector before applying SOM. In this study, we do not apply any filtering but use the original field to consider also the extreme values. As for the spatial domain, we used the WRF-Chem domain 2 (Fig. 1) in the SOM method. The description is added in Line 195 and Line 198

L210: remapping to a much finer resolution is not recommended, one cannot "create" the physical information not included in the coarser data. I suggest to remap WRF data to MERRA, and then validate.

R: Because of the relatively coarse resolution in MERRA-2 data and the small region we focus on, only 483 (23 x 21) grid cells can be used in the SOM training process. With very few samples, SOM may not accurately capture the representative spatial (transport) pattern. In this consideration, we bilinearly interpolated the data to a higher resolution with 110 x 110 grid cells (WRF-Chem outer domain) to increment the number of samples. The results after remapping have been examined. Below we show the surface dust

concentration and dust fluxes at meridional (v) and zonal (u) directions on the first day of each month in original MERRA2 and remapped data. It turns out that the remapped maps well match the original MERRA-2 maps.

[Figure]

Fig. R9 The low-level dust concentration (ug m$^{-3}$) and meridional and zonal dust fluxes (ug m$^{-2}$ s$^{-1}$) on the first day of each month.

L222: this is not evident from Fig. 2, can you provide a ref?

R: A reference has been added. Thank you.

Fig. 2: please provide uniform colorbars, for better comparison.

R: As discussed in the manuscript, the magnitude of dust loading from MERRA-2 is much higher than that from WRF-Chem, yet their spatial distributions are similar. If we use the range 5000-25000 (ug m$^{-2}$; WRF-Chem colorbar), dust loading over most regions in MERRA-2 will exceed the maximum range and the spatial distribution of dust loading is undiscernible. Similar condition will be reached if we use the range 30000-50000 (ug m$^{-2}$; MERRA-2 colorbar).

Sec. 3.2: it would also be interesting to analyse the preferred transitions of circulation types along with persistence.

R: We investigated the transition between SOM types by selecting the timesteps (3-hourly model output) associated with a type and counting the frequency of each type that occurs on the following timestep. Such frequencies reflect the likelihood (expressed as a percent of timestep) that a type persists (if it is the same type) or transits (if it is a different type). As shown in Table R1, the highest counts/frequencies are found along the diagonal, reflecting strong persistence for all types. The transition probabilities between different types (off-diagonal counts) are small and insignificant, which indicates that different types tend to be unrelated to each other. The analyses of transitions and persistence of SOM types produce very interesting results, which are worth further analyses and could be extended in our future study.

Table R1. Contingency tables among the four SOM types from WRF-Chem 3 hourly data during February-June 2019. The respective transition probabilities (indicated in parentheses) are obtained by dividing separate type counts by the sum of the columns of each row.

| From/To | Type 1 | Type 2 | Type 3 | Type 4 |
|---|---|---|---|---|
| Type 1 | 390 (0.91) | 22 (0.05) | 16 (0.04) | 1 (0.00) |
| Type 2 | 15 (0.05) | 253 (0.87) | 2 (0.01) | 19 (0.06) |
| Type 3 | 21 (0.10) | 2 (0.01) | 177 (0.83) | 13 (0.06) |
| Type 4 | 2 (0.01) | 13 (0.05) | 18 (0.07) | 235 (0.88) |

L274: how can we appreciate the dust deposition in Fig. 3?

R: The dust deposition pattern for each SOM type is shown in Figure 9. In the SOM clustering, we used a combination of 2D variables, including dust deposition, low-level meridional and zonal dust transport fluxes, and mid-level meridional and zonal dust transport fluxes. Therefore, the clustering results include the full information of dust deposition, low-level transport, and mid-level transport in each type.

L276: Fig. 6 is discussed before Fig. 5. Please change the order of figures (it also makes more sense to discuss local scale before teleconnections).

R: We agree. Figure 6 has been put forward.

Fig. 5: could you add some reanalysis aerosol product to display possible transport paths across the Pacific?

R: Per the reviewer's comment, we added dust optical depth (DOD) from MERRA-2 on Fig. R10 and Fig. 5. Below the dotted regions indicate DOD higher than 0.03. Dust emitted from East Asia can only reach 150°W and 180° in types 1 and type 2, respectively, while it reaches the North American west coast in types 3 and 4. The transport pathways

generally follow the isobars and wind patterns: In type 3, dust excurses north to Alaska and then travels south along the U.S. west coast. In contrast, dust from east Asia is transported zonally through the straight zonal isobars in type 4. The discussions are added in Lines 344-345.

"The MERRA-2 reanalysis DOD (Fig. 6c) further shows dust originating from Asia is transported towards North America following the isobars and wind patterns (discussed further in section 3.2.3)."

[Figure]

Fig. R10 Geopotential height (gpm) and wind vectors (m s-1) at 500 hPa in each of the four SOM types in WRF-Chem. The dotted regions indicate DOD higher than 0.03 from MERRA-2

L312: please provide a ref.

R: Revised as suggested.

L314: how can we appreciate the dust deposition in Fig. 3c? (see also the comment to L274)

R: Please see our response to your comment before.

L342: showing some reanalysis aerosol product could also be helpful in visualising the transport pattern.

R: We agree. Below we add the dust optical depth from MERRA-2 during the type 3 (NPH-related) and type 4 (Cross-Pacific Zonal transport) events. The transport pathway is more evident in the reanalyses data because of fewer missing values. In both types, we find dust emitted from east Asia is transported zonally to 150 °W. In Type 3, dust then

excurses meridionally to Alaska and then south along the U.S. west coast as there exists high pressure over the east Pacific. In contrast, dust from east Asia is consistently transported eastward to the West US coast through the straight zonal isobars at the middle level in type 4. We add the corresponding discussions in Lines 356-359 and Lines 384-385

"The dust transport pathway shows that after being emitted from East Asia and the Gobi Desert, dust is transported zonally to 150 °W, excursing north into Alaska/Canada and then traveling south along the U.S. west coast. Similar conclusions can be drawn with more evident pathways using DOD from MERRA-2 reanalyses (Fig. 8e)."

"The DOD from MERRA-2 confirms the zonal pathways of dust transport with a smaller magnitude (Fig. 8d)."

[Figure]

Fig R11 (a) IR DOD from IASI and 500 hPa winds (m s$^{-1}$) from ERA5 over the North Pacific for a typical Type 3 case averaged between 2019-05-02 to 2019-05-08; (b) same as (a) but for a typical type 4 event averaged between 2019-04-01 to 2019-04-09; (c) DOD from MERRA-2 and 500 hPa winds (m s$^{-1}$) from ERA5 for a 3 event; (d) same as (c) but for a type 4 event; (e) latitude-height cross-section of aerosol species from CALIOP on 2019-05-08 (Type 3); (f) same as (e) but for a typical Type 4 case on 2019-04-09

Fig. 11 and 12: please use the same colorbar as Fig. 3, for better comparison. In Fig. 11, the relative importance of the Mojave desert looks reduced in MERRA, any comment?

R: We tried to use the same colorbar for Figs. 11-12 as Fig. 3. However, it turned out that the dust concentrations in MERRA-2 are 2-3 times of those in WRF-Chem (Figure 2), which makes plotting them with one colorbar inappropriate. Most regions in Fig. 12 have larger dust concentrations than the maximum value we used in Fig. 3b (> 2 ug m$^{-3}$). We also tried a wider range colorbar to include values in both WRF-Chem and MERRA-2 but found the spatial heterogeneity is too low and the emission source is undiscernible in WRF-Chem plot.

L465: can you please show and comment the interannual variability of the occurrence of the SOM clusters?

R: Please see our response to your major comments #2.

**Technical corrections**

L24: "We find that dust…"

R: Corrected as suggested.

L26: "mostly in winter and…"

R: Done. Thank you

L28: "in the presence of the NPH".

R: Revised as suggested

**Reviewer #2**

This study integrates the WRF-Chem simulations with satellite data and reanalysis products in sppring to understand: (a) where does the dust deposition on the Sierra Nevada come from? (b) how is the dust transported to the mountainous regions? and (c) how is the dust deposited on the Sierra Nevada? The self-organizing map (SOM) analysis is applied to identify four major clusters of dust sources. Dust deposition onto the Sierra Nevada is believed to be an important factor that will affect the snow melt in the region and influence the water resource in the American West. Results from this study are an important contribution to the study of dust-cryosphere interactions. The paper is generally well written. I recommend the paper be published in ACP after addressing some minor issues below.

We greatly appreciate the reviewer's positive and constructive comments to improve the clarification. Below we provide a point-by-point response to reviewer's comments. In the following paragraphs, the reviewer's comments are in black font and our responses are in blue.

1. in the abstract, it is necessary to say which data have been used in this study.

R: We use 1-year WRF-Chem output (dust fluxes and deposition), and 21-year MERRA2 dust output (dust fluxes and deposition) in this study. The data is mentioned in the revised abstract.

2. lines 116-118: please explain why you didn't use outputs from full simulations (September 20, 2018 to August 2021)?

R: As shown in Fig. R1 and many previous studies (Hand et al., 2016; Kim et al., 2021; Achakulwisut et al., 2017), higher dust concentrations mainly occur during March-May while large depositions are concentrated in March-June. We also include February which has moderate deposition and concentration. Months after June are excluded because there is not much snow on the Sierra Nevada and therefore the depositions after June have small impact on snow impurities. We add the above explanation in Lines 118-120

"The simulation period ranged from September 20, 2018, to August 31, 2019 while we only used output from February to June in consideration of both dust emission season and mountain snow existence (Hand et al., 2016; Kim et al., 2021; Achakulwisut et al., 2017)."

[Figure]

Figure R1: Monthly mean dust concentration and deposition over California from 2011-2021 MERRA2

3. lines 137-140: How does WRF-Chem define PM10? Is it defined with geometric size or aerodynamic size?

R: WRF-Chem defines dust size bins based on geometric size (the same way as in MERRA2), while PM10 is defined based on aerodynamic size. We realize that they are different and has revised the sentence as follow (Line 141):

"MERRA-2 simulates dust with diameter bins of 0.2–2.0 (DU001), 2.0–3.6 (DU002), 3.6–6.0 (DU003), 6.0–12.0 (DU004), and 12.0–20.0 (DU005) µm, while the MOSAIC 4-bin in WRF-Chem simulates dust with **geometric** size bins of 0.039–0.156, 0.156–0.625, 0.625–2.5, and 2.5–10.0 µm. We therefore use the dust concentrations of the first 4 size bins in MERRA-2 (DU001 + DU002 + DU003 + 0.74 * DU004) to match with dust concentration with **geometric size less than 10.0 µm** in WRF-Chem (https://gmao.gsfc.nasa.gov/reanalysis/MERRA-2/FAQ/)"

4. lines 230-232 (and throughout the paper): please consider to use alternative words for "underestimates" and "underestimation". It is kind of weired when one says that observations "underestimate" something with a reference to model simulations.

R: We agree that it is inappropriate to say that the observations underestimate model simulations. We have replaced the word with "lower" and "smaller" when comparing satellite retrievals with model results

5. Figure 2: Given that (c) and (d) are for dust optical depth, I would suggest that they also show dust optical depth ion (a) and (b). Also there is clear difference between MERRA-2 and MIDAS in terms of spatial pattern. Please explain the difference.

R: While the total AOD were computed in WRF-Chem, the optical depth from each aerosol specie was not separated. In this consideration, we have to compare dust loading in WRF-Chem and MERRA2. The comparison between simulated dust loading/transport and observed DOD/transport aims to show the common dust emission regions in both model and satellite observations rather than match them quantitatively.

The difference between MERRA-2 and MIDAS mainly lies in southeast part of our study region (Figs. 2b, d). MERRA-2 shows large dust loading in the Mojave Desert, consistent with IASI IR DOD, while MIDAS shows generally small DOD. The notable underestimation (compared with IASI retrievals) might come from uncertainty in MODIS AOD, as shown in Fig. R2. We did not find much literature comparing the MODIS retrieved AOD and site observations over the Western US deserts, yet Tao et al. (2017) reported that the underestimation (compared to ground observations) is prevalent over the extensive deserts in East Asia, mainly due to overestimation of surface reflectance in these regions. We briefly discuss the uncertainties in Lines 246-248

"MIDAS is reported to underestimate DOD from the Mojave Desert compared to AERONET DOD, which might be caused by the lower dust amounts simulated in MERRA-

2 (Gkikas et al., 2021) and the underestimation of MODIS AOD over the deserts as compared to ground observations (Tao et al., 2017)."

[Figure]

Figure R2 Observed visible DOD at the wavelength of 550 µm from MIDAS.

6. Figure 3 caption: "Figure 5" should be "Figure 6", I guess.

R: Sorry for the typo. We put Figure 6 forward in the revised version. Now the caption is correct.

7. Figure 10: These violin plots are not straightforward or even confusing. It was very hard for me to figure it out and understand. Is it possible to change it to a more conventional PDF line plots?

R: The violin plots has been replaced by conventional PDF line plots. Thank you.

[revised manuscript text omitted]

---

## Author Response (AR2)

**Reviewer #1**

I thanks the Authors for considering and responding to all my comments and suggestions, and I appreciate the work they did to improve the manuscript. In my opinion, results persented in the paper are now more robust and I defenetly recommend it for publication in ACP. Just after a few typos here and there are corrected (see e.g. L187, L344, L420).

R: We thank the reviewer for the positive and constructive comments to improve the clarification. We have corrected the typos mentioned above and check the manuscript thoroughly.

L187: The repeated phrase "Before the machine-learning process" has been deleted

L344: quotation mark has been deleted

L420: "Dust" has been corrected to "dust".